# Recruitment of DNA Repair MRN Complex by Intrinsically Disordered Protein Domain Fused to Cas9 Improves Efficiency of CRISPR-Mediated Genome Editing

**DOI:** 10.3390/biom9100584

**Published:** 2019-10-08

**Authors:** Nina Reuven, Julia Adler, Karin Broennimann, Nadav Myers, Yosef Shaul

**Affiliations:** Department of Molecular Genetics, Weizmann Institute of Science, Rehovot 76100, Israel; julia.adler@weizmann.ac.il (J.A.); karin.Broennimann@weizmann.ac.il (K.B.); nadav.myers@weizmann.ac.il (N.M.)

**Keywords:** Gene targeting, CRISPR/Cas9, homology-directed repair, genome editing, endogenous mutagenesis in cell lines

## Abstract

CRISPR/Cas9 is a powerful tool for genome editing in cells and organisms. Nevertheless, introducing directed templated changes by homology-directed repair (HDR) requires the cellular DNA repair machinery, such as the MRN complex (Mre11/Rad50/Nbs1). To improve the process, we tailored chimeric constructs of Cas9, in which SpCas9 was fused at its N- or C-terminus to a 126aa intrinsically disordered domain from HSV-1 alkaline nuclease (UL12) that recruits the MRN complex. The chimeric Cas9 constructs were two times more efficient in homology-directed editing of endogenous loci in tissue culture cells. This effect was dependent upon the MRN-recruiting activity of the domain and required lower amounts of the chimeric Cas9 in comparison with unmodified Cas9. The new constructs improved the yield of edited cells when making endogenous point mutations or inserting small tags encoded by oligonucleotide donor DNA (ssODN), and also with larger insertions encoded by plasmid DNA donor templates. Improved editing was achieved with both transfected plasmid-encoded Cas9 constructs as well as recombinant Cas9 protein transfected as ribonucleoprotein complexes. Our strategy was highly efficient in restoring a genetic defect in a cell line, exemplifying the possible implementation of our strategy in gene therapy. These constructs provide a simple approach to improve directed editing.

## 1. Introduction

CRISPR (clustered regularly interspaced short palindromic repeat) and Cas (CRISPR-associated) proteins are part of the RNA-based adaptive immune system in bacteria and archaea [1]. This system has been adapted to create simplified tools for site-specific cleavage of genomic DNA in a wide variety of organisms [2,3,4,5,6]. Cas9 is a DNA endonuclease, which is targeted to a specific target site by an RNA guide sequence complementary to the target sequence. In the simplified tool version, a single guide RNA (sgRNA) is used in which the CRISPR RNA (crRNA) with homology to the target site is fused in a chimeric construct to the trans-activating crRNA (tracrRNA) [7]. The CRISPR/Cas9-mediated double-strand break (DSB) is usually repaired by the endogenous cell repair machinery, either by non-homologous end joining (NHEJ) or homologous recombination (HR), with NHEJ acting as the predominant repair pathway. NHEJ is highly efficient but error prone, and produces small insertions or deletions (indels), generally resulting in frame-shift mutations, which can effectively inactivate a gene (gene knock-out). If we provide a donor template DNA with homologous regions (“homology arms”) flanking the targeted site, homology-directed repair (HDR) can occur, producing edited sites with specific, targeted modifications (gene knock-in).

Recently, several groups have reported strategies to increase HDR by reducing NHEJ, stimulating HR, or by restricting the Cas9 cleavage to the portion of the cell cycle with optimal HDR. These approaches have included inhibiting NHEJ by knockdown of the factors involved or by inhibiting ligase IV pharmacologically [8,9]; approaches have also included pharmacologically enhancing HDR [10] or by restricting Cas9 expression to the S/G2/M portion of the cell cycle by fusing Cas9 to a domain of geminin [11]. Precise control over the timing of Cas9 expression can also be achieved by using photoactivatable Cas9 [12]. Improving the HDR/NHEJ ratio is critical for clinical applications, and application of these methods, such as restricting Cas9 activity to the relevant portion of the cell cycle, does show promise in the editing of human stem and progenitor cells [13]. Altering the global DNA repair environment in the cell has the potential danger of introducing unwanted mutations at random sites. As such, modified Cas9 constructs have been developed that fuse functional repair domains or repair protein recruiting domains to the Cas9 nuclease. One study has used the fusion of a 296 aa domain of the repair protein CtIP to Cas9, and has achieved increases in editing of 1.5–2.5 times over the unmodified Cas9 [14]. A similar approach using fusion of Cas9 to CtIP, Mre11, or Rad52 showed two times the improvement in HDR as well as a significant reduction in NHEJ [15].

Due to their short generation time and high copy numbers, viruses have evolved highly efficient means to replicate their genomes: encoding their own replication proteins and evolving effective ways of interacting with the host cell machineries. HSV-1 encodes a two subunit recombinase, consisting of a 5’-3’ exonuclease, UL12, and a single stranded binding/pairing protein, ICP8. The two proteins together can mediate recombination (strand exchange) in vitro [16,17]. In vivo, expression of the recombinase, and even expression of UL12 alone increases recombination by the single-strand annealing pathway and inhibits NHEJ [18]. Importantly, the N-terminus of UL12 has been shown to efficiently recruit the cellular MRN complex (Mre11/Rad50/Nbs1) [19]. The MRN complex coordinates double-strand break (DSB) repair, resects dsDNA ends, and recruits factors needed for HDR, as reviewed in [20]. Here, we show that fusion of the N-terminal 126 amino acid domain of UL12 to Cas9 promoted the association of the chimeric Cas9 protein with the cellular MRN complex. The chimeric Cas9 constructs were generally two times more efficient in promoting specific editing of genomic loci and highly efficient in correcting a genomic point mutation. These chimeric Cas9 constructs provide a simple and versatile means for improving CRISPR-mediated genome editing.

## 2. Materials and Methods

### 2.1. Cells and Cell Culture

Human embryonic kidney cells HEK293FT (ThermoFisher) and HEK293, HCT116, and HeLa (all ATCC) cells were grown at 37 °C in a humidified incubator with 5.6% CO_2_ in Dulbecco’s modified Eagle’s medium (DMEM; GIBCO, Life Technologies, Thermo Scientific, Waltham, MA, USA) supplemented with 8% fetal bovine serum (GIBCO), 100 units/mL penicillin, and 100 µg/mL streptomycin. BHK21/13 ts13 cells [21] were grown under the same conditions but at 32 °C. The restrictive temperature used for BHK21 ts13 and the HEK293 TAF1ts cells (this study) was 39.5 °C. Puromycin was from GoldBio (St. Louis, MO, USA). Light microscopy photographs of cells were performed using an Olympus (Tokyo, Japan) IX70 microscope connected to a DVC camera. The XTT assay (Biological Industries) was used to quantify cell proliferation and viability.

### 2.2. Plasmids and Transfection

The SpCas9/sgRNA expression plasmids were based on pX330-U6-Chimeric_BB-CBh-hSpCas9, a gift from Feng Zhang (Addgene plasmid # 42230; http://n2t.net/addgene:42230; RRID:Addgene_42230) [2], and pU6-(BbsI)_CBh-Cas9-T2A-mCherry (a gift from Ralf Kuehn (Addgene plasmid # 64324) [8]). These vectors express both the active Cas9 (3xFlag-tagged, and with nuclear localization signal (NLS) sequences at the N- and C-termini of Cas9), under control of a CBh promoter, and the sgRNAs, under control of the U6 promoter. We modified the site for insertion of the guide RNA sequences, using BsaI sites instead of BbsI for cleavage, but preserved the same sticky ends for cloning guide sequences as the original plasmids. The sequence encoding the first 126 aa of HSV-1 UL12 was amplified from pSAKUL12/12.5 [22] using the primer pairs and restriction sites listed in Appendix A, and cloned into pX330 or pU6-(BbsI)_CBh-Cas9-T2A-mCherry. In the Cas9-T2A-mCherry constructs, the fusion of the UL12 fragments to the C-terminus of Cas9 was upstream of the T2A peptide sequence. Guide RNA and ssODN sequences as well as other primers used for PCR are listed in Appendix A. Plasmid donor DNA constructs used pBlueScript KS- as a backbone. The homology arm DNA was amplified by PCR from the cell lines’ genomic DNA and cloned into the backbone using the restriction sites noted in Appendix A. The sequence for YFP was amplified from pSYFP2-C1, a gift from Dorus Gadella (Addgene plasmid # 22878) [23]. The sequence for the puromycin resistance gene was amplified from pEFIRES [24]. Transfections were done by the calcium phosphate method as described in [25], JetPEI^®^ (Polyplus-transfection SA, Illkirch, France), or with polyethylenimine (PEI) 25K (Polysciences) prepared at 1 mg/mL and used similarly to the commercial JetPEI reagent. Transfection of ribonucleoprotein complexes (RNP) was done with Lipofectamine CRISPRMax (ThermoFisher). sgRNA for RNP transfection was in vitro transcribed using MEGAshortscript T7 (Ambion) from PCR amplicons made using the primers listed in Appendix A. The template for this PCR was the pX330-based vector encoding the sgRNA. Bio-Tri (BioLabs) was used to purify the RNA.

### 2.3. Immunoblot and Coimmunoprecipitation

Immunoblots and immunoprecipitations (IPs) were performed as previously described [25] using RIPA buffer (50mM Tris-HCl pH 7.5, 150mM NaCl, 1% Nonidet P-40 (*v*/*v*), 0.5% deoxycholate (*v*/*v*), 0.1% SDS (*w*/*v*)) supplemented with cocktails of protease inhibitors and serine/threonine and tyrosine phosphatase inhibitors (Apex Bio). Antibodies used were monoclonal anti-β-tubulin, anti-β-actin, and anti-FLAG M2 (Sigma, St. Louis, MO); anti-Mre11 (18), anti-Rad50 (G-2), anti-Nbs1 (B-5), anti-PSMB6 (B-6) (Santa Cruz Biotechnology, Santa Cruz, CA, USA); anti-p73 BL906 (Bethyl Laboratories, Montgomery, TX, USA). The polyclonal Living Colors antibody (Clontech) was used to detect SYFP, and detection of these proteins was more sensitive if the samples were not boiled prior to separation by SDS-PAGE. For immunoprecipitation (IP) of Flag-tagged proteins, anti-Flag M2 agarose (Sigma) was used. Horseradish peroxidase-conjugated secondary antibodies were from Jackson ImmunoResearch Laboratories, West Grove, PA. Enhanced chemiluminescence was performed with the EZ-ECL kit (Biological Industries, Kibbutz Beit Haemek, Israel) and signals were detected by the ImageQuant LAS 4000 (GE Healthcare, Piscataway, NJ, USA). Intensities of bands were quantified by the ImageQuant TL software. For comparison of multiple experiments, values within one experiment were normalized to a standard set at 1. “SEM” refers to standard error of the mean.

### 2.4. Statistical Analysis

A student’s t-test (two-sided, unequal variance) was performed to assess significance.

### 2.5. Flow Cytometry

Live cells were harvested and washed with PBS. For each measurement, 30,000–100,000 cells were collected by the BD LSRII flow cytometer (Becton Dickinson, Mountain View, CA, USA) and analyzed with the BD FACSDiva software (BD Biosciences).

### 2.6. Cell Staining

Cells were washed with phosphate-buffered saline (PBS) and fixed with ice-cold methanol for 10 min. Cells were then briefly stained with 0.1% crystal violet in 25% methanol, at room temperature, then rinsed with double distilled water until stain no longer leached into wash water. Crystal violet-stained plates were scanned with a desktop scanner (at a resolution of 300 dpi). ColonyArea, a java-based plugin for ImageJ, was used to quantify colony area and intensity of each well [26].

### 2.7. Recombinant Cas9 Production and Purification

pET-28b-Cas9-His was a gift from Alex Schier (Addgene plasmid # 47327; http://n2t.net/addgene:47327; RRID:Addgene_47327) [27] and was modified to contain N-terminal SV40 NLS and C-terminal Nucleoplasmin NLS. Two tags were fused in frame with Cas9: 3xFLAG tag at the N-terminus and His_6_ at the C-terminus of the protein. The resulting construct was used as a control and served as the template in which the UL12 sequences (126 AA) were introduced upstream or downstream to Cas9. Engineering of all constructs were performed either by RF cloning [28] or by Inverse PCR [29]. The resulting vectors were transformed into *E. coli* Rosetta 2 pLys-S competent cells. Proteins were purified by immobilized metal ion chromatography (IMAC) using a HiTrap *FF_*5 mL cartridge (GE Healthcare) using an FPLC system (ÄKTA GE Healthcare Life Sciences). Details of the procedure are listed in Appendix A.

## 3. Results

Homology-directed genome editing using the CRISPR/Cas9 system is hampered by the reliance on the cellular DNA repair machinery, which usually favors NHEJ over HR. To improve this editing, we looked for a way to localize the needed repair factors to the site of the break. We surmised that fusion of Cas9 to a domain known to recruit DNA repair factors would help to improve editing of desired sites, without affecting global DNA repair activity in the cell. To this end, we chose the HSV-1 alkaline nuclease, UL12. The N-terminal 126 amino acid domain of UL12 recruits the cellular MRN complex that is necessary for homology-directed repair [19]. Analysis of the UL12 protein by IUPred2A [30], as seen in Figure 1A, demonstrated that the N-terminal 126aa domain is intrinsically disordered; in contrast, the C-terminal catalytic domain of the protein is mostly structured. IUPred2A is a web interface that identifies disordered protein regions using IUPred2 and disordered binding regions using ANCHOR. The 126aa domain of UL12 had high scores for intrinsic disorder and for predicted binding regions. Intrinsically disordered protein domains, owing to their flexibility, often function as protein–protein interaction hubs, as reviewed in [31]. For these reasons, we chose this compact and flexible domain to recruit DNA repair complexes to the site of the double-strand break created by the Cas9 nuclease. Furthermore, this domain does not possess the nuclease activity, as it is dispensable for the nuclease activity of UL12 [32,33] and is not part of the seven conserved domains of herpesvirus alkaline nuclease homologs [34]. We created chimeric constructs, with the UL12 disordered 126aa region fused to the N- or C- termini of SpCas9, as seen in Figure 1B. 

To test whether the chimeric constructs interact with the endogenous MRN complex (Mre11, Rad50, and Nbs1), we transfected HEK293 cells with the Cas9 plasmids encoding a non-specific guide RNA that is not expected to induce cleavage of the DNA. This was to insure that any interaction was through the Cas9 constructs and not mediated by cleaved DNA. The endogenous MRN complex co-immunoprecipitated with the U_N_ and U_C_ constructs but not with the wt (wild-type) Cas9, as seen in Figure 1C. This demonstrated that the UL12 aa1-126 disordered fragment, when fused to Cas9 at either the N- or C-terminus, promoted efficient association with MRN.

We first tested the MRN recruitment strategy in making a small insertion at a precise location into an endogenous gene. To this end, we used CRISPR editing of HEK293 cells to add a Flag tag to the C-terminus of proteasome subunit β1, encoded by the PSMB6 gene, as seen in Figure 2A. HEK293 cells were transfected with the Cas9/sgRNA plasmids and ssODN, and the pool of transfected cells was harvested for analysis by SDS-PAGE and immunoblotting. We quantified the PSMB6-Flag level in the transfected cells as a measurement of editing success. Success in generating the chimeric PSMB6-Flag with wt Cas9 was dose-dependent and a maximal level was obtained with the highest Cas9 levels. In contrast, we obtained maximal PSMB6-Flag level with 20 ng of the U_N_ Cas9 plasmid, which is about five times lower than the amount of naïve Cas9 plasmid needed for maximal editing, as seen in Figure 2B. Furthermore, the maximal level obtained under MRN recruitment, using U_N_ Cas9, was higher than that obtained with the maximal level of wt Cas9, indicating that high levels of editing were achieved with very low levels of the chimeric Cas9. These data suggest that recruitment of the MRN complex dramatically potentiates HDR-dependent targeted editing.

To further test the efficiency of the chimeric constructs, we tested the insertion of a large fragment, using a plasmid donor template. Using the same guide targeting PSMB6, we fused SYFP to the C-terminus of proteasome subunit β1, as seen in Figure 3A. We previously used and characterized this system, which produces a 50 kDa PSMB6-YFP fusion protein that is incorporated into the endogenous proteasomes, as we have seen by native gel analysis, as seen in Appendix A [35]. PCR analysis of two single-cell purified clones suggest bialleic insertion of the cassette, as seen in Appendix A. Using this system, both the U_N_ and U_C_ constructs produced an increase in editing when compared with the wt Cas9 seen in both HCT116, as seen in Figure 3B, and HEK293, as seen in Appendix A. Quantification of editing in HEK293FT by FACS analysis also showed improved editing by the chimeric constructs over wt Cas9, as seen in Figure 3C and Appendix A. The FACS results showed that 1–3% of the total cell population was edited. Yet, the same double increase with the MRN-recruiting Cas9 constructs was observed here that was seen with western blot analysis. To determine whether the level of editing we obtained was limited by the amount of donor plasmid, we titrated the PSMB6-YFP donor plasmid, as seen in Figure 3D. Interestingly, increasing the amount of donor plasmid did not increase the editing level, indicating that the donor was not the limiting factor, although there did appear to be a threshold of maximum editing in all these experiments. 

Delivery of Cas9 by protein transfection can be advantageous for several reasons discussed below, and thus we purified recombinant constructs of the wt, U_N,_ and U_C_ Cas9, as seen in Appendix A. All three proteins were expressed, but the wt and U_C_ were expressed to a higher level than the U_N_ and were more highly purified. Editing of PSMB6-YFP by the recombinant wt and U_C_ Cas9, and in vitro transcribed sgRNA, showed a 3-fold improvement in editing by the U_C_ Cas9, as seen in Figure 3E.

To test whether the chimeric constructs showed increased editing efficiency at other loci, we targeted the p73 locus to generate a YFP-p73 fusion under control of the endogenous p73 promoter. In the 1.4-kb cassette, we included a puromycin resistance gene (*pac*, puromycin N-acetyl transferase), followed by a T2A peptide, the SYFP gene and a short linker, as seen in Figure 3F. Expression of the cassette, driven by the endogenous p73 promoter, should generate puromycin resistance and expression of the YFP-p73 fusion protein. Using this system, we generated the expected YFP-p73 construct, as demonstrated by western blot analysis of two single-cell-purified clones, one originating from transfection with wt Cas9 and one from U_N_, as seen in Figure 3G. Both clones show expression of a 100 kDa protein that was recognized by both the anti-p73 and anti-YFP antibodies. Wild type p73 was detected in the naive sample but not in the edited clones. PCR analysis of the clones show that both clones have insertion of the cassette. The clone generated by U_N_ Cas9 appears to have biallelic insertion of the cassette, whereas the clone generated by wt Cas9 appears heterozygous, as seen in Appendix A. To compare editing efficiencies of the Cas9 constructs, we used puromycin resistance as a readout for editing. We used the XTT assay to quantify cell survival after puromycin treatment. The U_N_ and U_C_ chimeric Cas9 constructs were twice as efficient as the wt Cas9 in editing, as seen in Figure 3H. The XTT assay results comparing growth of cells with and without puromycin showed that the editing efficiency was about 10%, as seen in Appendix A. The pools of the puromycin-selected cells all expressed the YFP-p73 fusion protein, as seen in Appendix A, indicating that the cassette was faithfully inserted. 

To provide further evidence for the importance of MRN recruitment in increasing HDR efficiency in genetic editing, we generated truncation mutants of the disordered 126aa region fused to Cas9. This region has a consensus NLS (nuclear localization signal) at aa35–39 [22,36] and the aa1–50 UL12 fragment demonstrated NLS activity when fused to SYFP, as seen in Appendix A. However, the 1–50 UL12 fragment when fused to the N-terminus of Cas9 (U_NLS_ construct) did not improve editing, as seen in Figure 3I,J. Furthermore, it has been reported that deletion of the first 50aa does not hamper MRN recruiting activity [19]. Thus, we wished to determine whether the 50–126aa fragment is sufficient for MRN recruitment in the context of the chimeric constructs. We therefore tested fusion of this truncated domain to Cas9 at both the N- (U_Ns_ construct) and C-termini (U_Cs_ construct). The U_Ns_ and U_Cs_ showed low or no effect on editing efficiency over the naive Cas9, as seen in Figure 3I,J. The endogenous MRN complex (Mre11, Rad50, and Nbs1) co-immunoprecipitated with the disordered 126aa fused fragment (U_N_ and U_C_ constructs) but not with the U_Ns_ and U_Cs_ shortened chimeric constructs, as seen in Appendix A. Taken together, these results show that fusion of the MRN-recruiting domain to Cas9 correlates with improved HDR-dependent genome editing.

Next, we looked for a model of a genetic disease that is caused by a single amino acid change. To this end, we looked for a temperature-sensitive (ts) cell line, and used CRISPR editing to revert the ts mutation to the wild type sequence. We used BHK21/13 ts13, with a well-defined point mutation, G690D, in the TAF_II_-250 (TAF1) gene on the X chromosome, which encodes the largest component of the basal transcription complex TFIID [37]. These cells, a mutant isolate of BHK (baby hamster kidney) cells, grow at permissive temperature (we used 32 °C), but die when incubated for several days at the restrictive temperature of 39.5 °C. As seen in Figure 4A, CRISPR editing of this locus produced colonies of the edited cells that were no longer temperature-sensitive. The chimeric constructs showed an advantage in editing efficiency, with three to four times more editing seen with U_N_ and U_C_, as seen in Figure 4B.

The human and hamster TAF1 genes are highly homologous; accordingly, we used CRISPR editing in HEK293 cells to make the same mutation that was identified in the BHK21/13 ts13, as seen in Appendix A. In human cells, this is TAF1 G716D, in exon 13 of the TAF1 gene on the X chromosome. HEK293 have three X chromosomes (https://www.atcc.org/products/all/crl-1573.aspx#characteristics), and the clones we isolated each had one allele with the ts mutation, and different insertions or deletions (indels) in the other two alleles, as seen in Appendix A. Similar to the ts13 cells, the HEK293 TAF1ts cells grew at the permissive temperature (here 37 °C), but did not survive at the restrictive temperature (39.5 °C). Following transfection of the cells with Cas9/sgRNA and ssODN, cells were transferred to 39.5 °C, and colonies of surviving cells were stained and quantified. When non-specific sgRNA or ssODN were used, no HEK293 TAF1ts cell growth was observed, suggesting spontaneous reversion is a very rare event if any, as seen in Figure 4C1–3. Similarly to what we observed with the ts13 cells, the U_N_ and U_C_ constructs were, on average, twice as efficient in editing of the HEK293 TAF1ts site, as seen in Figure 4C,D. The larger size of the colonies produced by editing with the U_N_ and U_C_ constructs may indicate faster editing, and thus an earlier start in colony formation in these plates. The reversion of the mutation was specific to the CRISPR/Cas9 mediated HDR, and occurred only with expression of the ts-specific sgRNA and the TAF1 ts-correcting template ssODN together, as seen in Figure 4C. The levels of expression of the Cas9 constructs were the same, as shown by immunoblotting of samples of the transfected cells, as seen in Appendix A. These data suggest that MRN recruitment by the chimeric Cas9 constructs improved HDR-dependent CRISPR editing of a genetic disease model.

## 4. Discussion

Genome editing has been a long sought-after goal; with CRISPR/Cas9, this vision is becoming reality. However, stumbling blocks remain, which impair the efficiency of editing. Here, we demonstrate a simple approach to improve the yield of genomic edits that require HDR. Cas9 is a bacterial protein and, as such, is not “familiar” with the human cellular environment and incapable of recruiting important cellular machinery. We hypothesized that humanizing Cas9 to the level of recognizing the cellular recombination machinery would improve its editing capacity. To this end, we fused SpCas9 to a protein domain that would promote HDR locally at the break site to achieve more HDR without perturbing global DNA repair pathways. Our results show that the fusion of the N-terminal UL12 disordered 126aa fragment to Cas9 provided more efficient editing in a number of different systems when using both ssODNs and plasmid DNA as donor. Furthermore, maximal levels of editing were achieved with less of the chimeric Cas9 than the wt Cas9, which is advantageous in terms of economy, and also is predicted to reduce the chances for off-target effects. In our study, we focused on editing endogenous genes, selectively looking at the HDR output. A drawback to this approach is that we could not quantify the HDR/NHEJ ratio using these systems. 

Our approach of fusing a disordered fragment with recruiting activity but not enzymatic activity avoids problems that arise with overexpression of Cas9 with extra enzymatic activities. Recent studies employing fusion of base-editors to Cas9 have also shown that untargeted base-editing can occur [38]. Therefore, fusion of enzymes to Cas9 can help target the added activity to the site specified by the sgRNA but does not necessarily restrict the activity to the site. In the case of the MRN-recruiting domain of UL12, it is beneficial that this is a domain with recruiting activity but not enzymatic activity. Our data indicate that the MRN complex associated with the chimeric Cas9 constructs. However, the activation of the Mre11 resection activity and recruitment of other repair factors by MRN should occur only in the case of an actual double-strand break, mediated by the guide RNA-directed endonuclease activity of Cas9. 

The N-terminal 126aa of UL12 is an intrinsically disordered region with a high score for disordered binding regions as well, as analyzed by IUPred2A, as seen in Figure 1A [30]. Intrinsically disordered proteins (IDPs), whether completely disordered or possessing intrinsically disordered domains or regions, are highly enriched for protein–protein interaction motifs [39,40,41]. The lack of a fixed structure enables these proteins to make contacts with several partners, and thus are highly suited to serve regulatory roles as well as to serve as hubs for the recruitment of other proteins [31]. Proteins that interact with viruses are highly enriched in disorder, suggesting that viruses often interact with host proteins via intrinsically disordered domains [42,43]. The analysis of UL12 shows a high level of predicted disorder and binding regions in the 126aa N-terminal region, as seen in Figure 1. The recruitment of the MRN complex by the UL12 N-terminal domain is consistent with its IDP nature and, in the context of HSV-1 viral infection, it may use MRN recruitment to sites of viral replication compartments to promote DNA recombination that is associated with HSV-1 DNA replication [44]. Due to their relatively small size, fast generation times, and large numbers of progeny, viruses have typically developed, through evolutionary pressure, the most efficient mechanisms for virus–host interactions. Analysis of viral proteomes has shown many with a high level of intrinsic disorder [45]. Intrinsic disorder enables viruses to cope with many of the problems it faces, such as the need to exact multiple functions out of compact genomes and the need to interact with a variety of host proteins [42]. Thus, the small 126aa domain has been selected through evolution to efficiently recruit the host MRN complex that the virus requires. The IDP nature of the 126aa domain may also explain why it can be fused at either the N- or C- terminus of Cas9 without compromising its efficiency or the activity of the Cas9.

The small size of the MRN recruiting domain of UL12 has many potential benefits for CRISPR/Cas9 editing. Applications using AAV-derived vectors to deliver Cas9 are limited in their coding capacity (reviewed in [46]), and thus some of the other chimeric Cas9 proteins that have been developed recently may be less well suited to this application due to the large size of the constructs. In addition, fusion of the 126aa domain to Cas9 does not hamper production of recombinant Cas9, as seen in Appendix A, useful for protocols using delivery of ribonucleoprotein (RNP) complexes to cells. Our analysis showed that also with RNP delivery of the Cas9/sgRNA, the chimeric Cas9 performed better than wt Cas9, as seen in Figure 3E. Cas9 protein transfection has been shown to be an efficient means to engineer a variety of cell lines and the off-target effects are lower because of the faster depletion of the Cas9 protein in comparison with Cas9 expressed from plasmids [47]. 

Our results showed that lower levels of the chimeric Cas9 were sufficient to achieve maximal editing, and this is useful for several reasons. As noted above, expression of Cas9 from plasmids leads to high levels of Cas9 for relatively long periods of time. Editing using protein transfection [47] or strategies to restrict Cas9 expression [48] have achieved fewer off-target effects by limiting Cas9 levels. The chimeric Cas9 constructs were effective at much lower levels than the wt Cas9, and thus would be predicted to have fewer off-target effects, which is particularly relevant for clinical applications. In addition, for protocols using costly recombinant proteins, the ability to achieve the same result with less Cas9 could significantly lower costs. 

In the systems we have tested, there appeared to be a threshold of maximum editing in each case, which was not improved with the addition of more Cas9/guide, or donor DNA. These results may offer some insight into the mechanisms of CRISPR-mediated HDR, and what may actually be the rate-limiting steps in this process. The chimeric Cas9 constructs presented here were able to generally double the number of editing events, which may represent the upper limit of inducing HDR in otherwise unmanipulated cells. The homologous repair machinery is active during the S-G2 portion of the cell cycle (reviewed in [49]). By recruiting the MRN complex to the site of the Cas9-induced break, we succeeded in inducing homologous repair in cells where this repair pathway is less optimal but still possible, due to low levels of the necessary factors. Although relatively modest, this improved efficiency can still greatly reduce workload in terms of reducing the numbers of potential clones that must be screened when preparing cell lines with endogenous mutations. Furthermore, in clinical applications, a two-fold increase could provide the needed push to cross a therapeutically relevant threshold [50]. Furthermore, the low levels of Cas9, combined with a technique that does not inhibit global DNA repair pathways, mean that the edited cells should not harbor unwanted mutations due to off-target effects or those resulting from impaired DNA repair.

## 5. Conclusions

To improve homology-directed repair (HDR)-dependent genome editing, we fused the intrinsically disordered N-terminal 126aa region of HSV-1 UL12 to either the N- or C-terminus of SpCas9. These chimeric constructs recruited the endogenous MRN complex, necessary for HDR. The chimeric constructs improved genome editing by a factor of two over the wild-type constructs, and were effective at lower amounts by a factor of five than the wild-type constructs. This was true whether the Cas9 constructs were delivered by plasmid transfection or transfection of nucleoprotein complexes. The chimeric constructs were more efficient at editing endogenous point mutations or inserting small tags encoded by oligonucleotide donors or inserting larger cassettes encoded by plasmid donor DNA. These constructs provide a simple approach to improve directed editing.

## 6. Patents

International patent application PCT/IL2019/050707, “Systems and methods for increasing efficiency of genome editing”, covering the constructs described in this manuscript, was filed by N.R. and Y.S. June 25, 2019. 

## Figures and Tables

**Figure 1 biomolecules-09-00584-f001:**
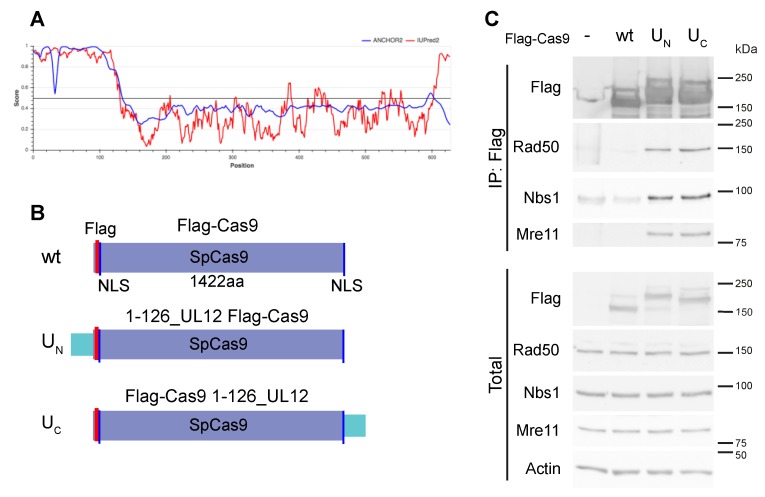
Intrinsically disordered region of HSV-1 UL12 fused to Cas9 recruits MRN complex. (**A**). Prediction of UL12 intrinsic disorder using the IUPred2A tool. The HSV-1 UL12 protein sequence was analyzed using IUPred2A, accessed 25.4.19 via https://iupred2a.elte.hu/. IUPred2A is a web interface that identifies disordered protein regions using IUPred2 and disordered binding regions using ANCHOR. The analysis of UL12 shows a high level of predicted disorder and binding regions in the 126aa N-terminal region. (**B**). Schematic representation of the SpCas9 constructs used. SpCas9 is indicated in purple; NLS, nuclear localization signal, indicated in blue; 126aa UL12 domain indicated in turquoise; Flag, 3xFlag, indicated in pink. (**C**). UL12 126aa domain fused to SpCas9 recruits cellular MRN complex. Co-immunoprecipitation (IP) of endogenous MRN complex with chimeric Cas9 constructs. HEK293 cells were transfected with Flag-mCherry (control) or the Flag-Cas9/guide-encoding plasmids indicated. The guide used was targeting TAF_II_250 in BHK ts13, seen in Figure 4A, with no predicted cleavage of human genomic sequences. Cells were harvested two days post-transfection and were subjected to IP with anti-FlagM2 agarose. IP and total cell extracts were analyzed by SDS-PAGE and immunoblot with the indicated antibodies. The Flag-mCherry control protein has a much smaller molecular weight (25 kDa) than the Flag-Cas9 proteins and is not shown in the figure.

**Figure 2 biomolecules-09-00584-f002:**
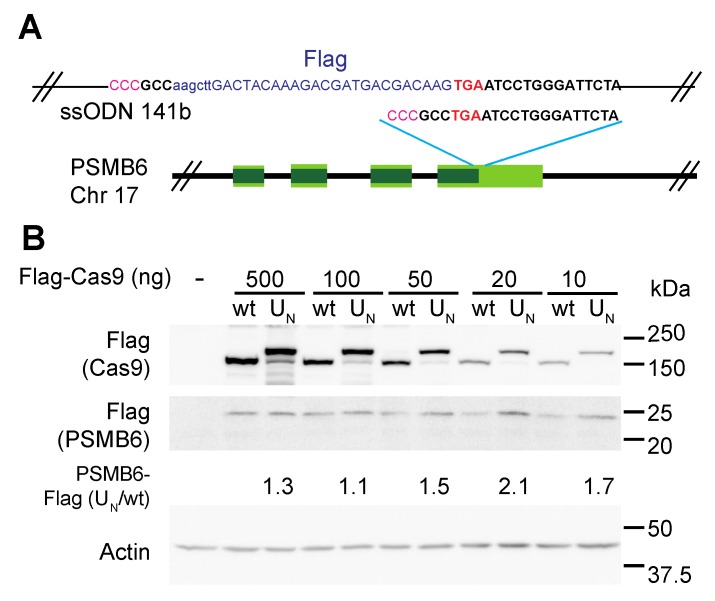
Tagging of endogenous gene is improved with MRN recruitment by chimeric Cas9. (**A**). Schematic representation of targeted PSMB6 locus. Stop codon is indicated in red. Guide sequence is in bold, with PAM sequence in pink (note that the guide is the minus strand). The region of the ssODN template encoding the Flag tag is shown. The left and right homology arms are 58 and 53 bp, respectively. (**B**). Insertion of Flag tag at C-terminus of PSMB6 is more efficient with MRN-recruiting Cas9 construct. HEK293 cells were transfected with the indicated amounts of Cas9/sgRNA encoding plasmids and with ssODN. Cells were analyzed by SDS-PAGE and immunoblotting with the indicated antibodies.

**Figure 3 biomolecules-09-00584-f003:**
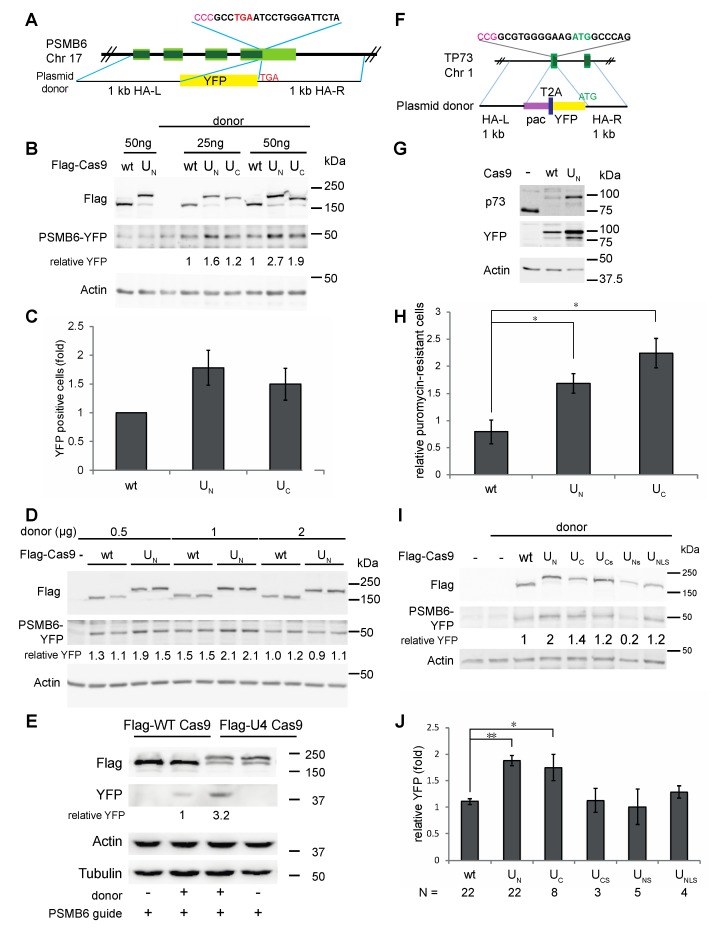
MRN recruitment by chimeric Cas9 improves insertion of large cassettes. (**A**). Schematic representation of targeted PSMB6 locus and donor plasmid. Stop codon is indicated in red. Guide sequence is in bold, with PAM sequence in pink (note that the guide is the minus strand). The region of the plasmid donor DNA with the 1 kb left and right homology arms (HA) and SYFP insert is shown. (**B**). Insertion of YFP tag at C-terminus of PSMB6 is more efficient with MRN-recruiting Cas9 constructs. HCT116 cells were transfected with the indicated amounts of Cas9/sgRNA encoding plasmids and with donor DNA. Amounts of total DNA in transfections were constant among samples, with empty plasmid (pBluescript) used to equalize amounts. Cells were analyzed by SDS-PAGE and immunoblotting with the indicated antibodies. (**C**). FACS analysis of PSMB6-YFP edited cells. HEK293FT cells were transfected with Cas9/sgRNA encoding plasmids and donor DNA. Cells were passaged twice and analyzed by FACS, with 100,000 cells analyzed per point. Summary of three independent experiments is shown. (**D**). Titration of donor DNA in PSMB6-YFP editing. HEK293 cells were transfected with the Cas9/sgRNA-expressing plasmids and the donor DNA indicated. (**E**). Recombinant chimeric Cas9 yields more edited cells than wt Cas9. HEK293 cells were transfected with 2 µg of recombinant Flag-Cas9 or U_C_ Flag-Cas9, 400 ng sgRNA, and 3.125 µg PSMB6-YFP donor plasmid, as indicated, using CRISPRMax reagent. Cells were harvested three days post-transfection, and analyzed by SDS-PAGE and immunoblotting with the indicated antibodies. Samples for this gel were not boiled to improve detection by the living colors antibody, which causes a slight change in migration of the 50 kDa PSMB6-YFP protein. (**F**). Schematic representation of targeted p73 locus and donor plasmid. Start codon is indicated in green. Guide sequence is in bold, with PAM sequence in pink (note that the guide is the minus strand). The region of the plasmid donor DNA with the 1 kb homology arms (HA) and SYFP insert is shown. (**G**). HEK293 cells were transfected with either wt or U_N_ Cas9/sgRNA constructs and the donor plasmid, encoding the pac-2A-YFP cassette flanked by 1kb homology arms. Cells were selected with 0.5 µg/mL puromycin and were plated as single cells to isolate clones. SDS-PAGE and immunoblot analysis is shown for two of these clones compared to naive cells. (**H**). Insertion of pac-2A-YFP cassette at p73 locus is more efficient with MRN-recruiting Cas9 constructs. HEK293 cells were transfected in biological triplicates with Cas9/sgRNA plasmids and donor DNA. Two days post-transfection cells were replated, with serial dilutions, into duplicate 96-well plates with or without 0.5 µg/mL puromycin. The XTT assay was used to quantify the relative amounts of puromycin-resistant cells, with the values from the plate without puromycin used for normalization. N = 3, * *p* < 0.023. (**I**). Editing of PSMB6-YFP using truncated chimeric constructs. HEK293 cells were transfected with Cas9/sgRNA plasmids and donor DNA. Cells were analyzed by SDS-PAGE and immunoblotting with the indicated antibodies. (**J**). Summary of editing of PSMB6-YFP by chimeric constructs. Quantification of multiple experiments using this system with the indicated wt and chimeric Cas9 constructs. In each experiment, the level of PSMB6-YFP signal obtained with one of the wt Cas9 samples was set to “1”, and the fold increase/decrease of the other samples was compared to this standard (** *p* < 10^−7^, * *p* < 0.04).

**Figure 4 biomolecules-09-00584-f004:**
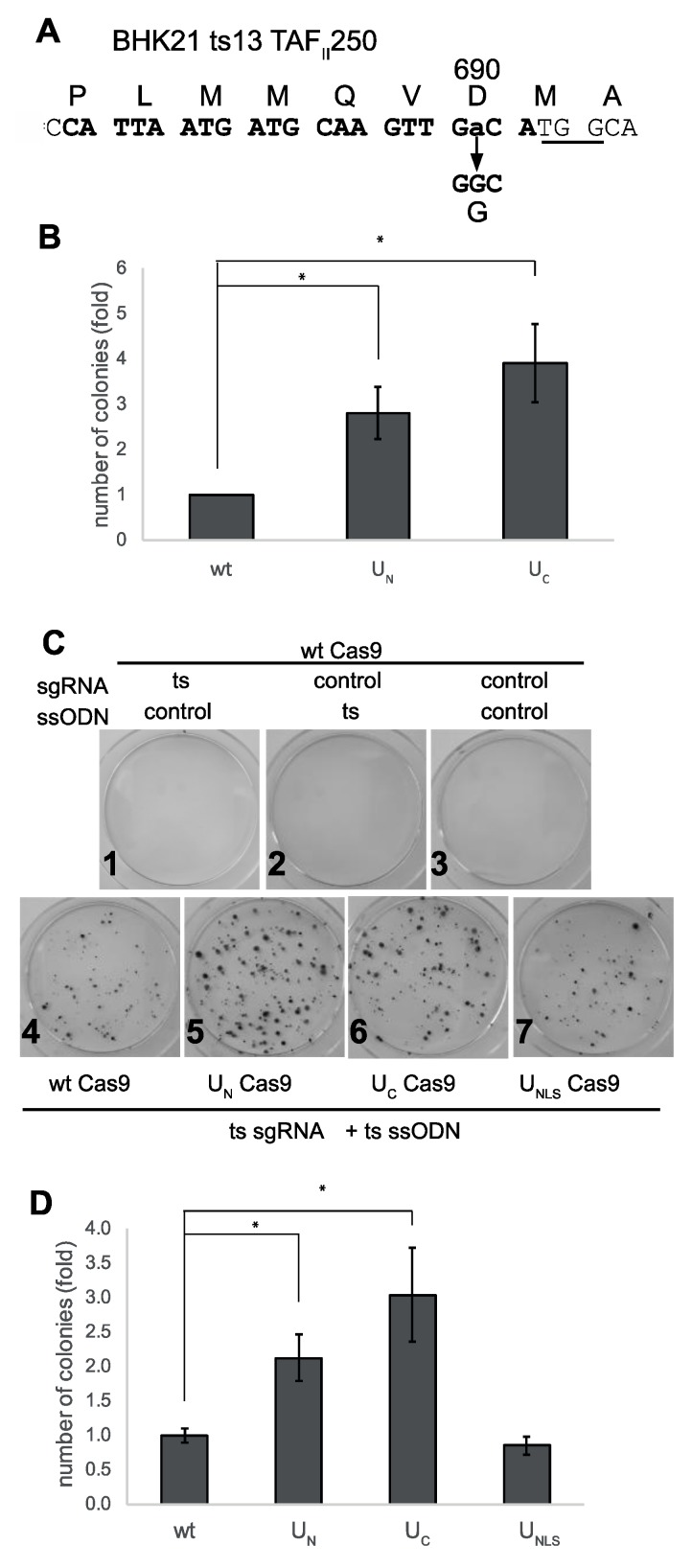
MRN recruitment improves editing of endogenous point mutation. (**A**). Site of the G690D ts mutation in BHK21 ts13 cells. The guide sequence targeting the mutation site is shown in bold, with the PAM site underlined. The ssODN used had 50bp homology arms. (**B**). Editing of the ts mutation in BHK21 ts13 is three to four times more efficient with MRN-recruiting constructs. BHK21 ts13 cells were transfected with the indicated Cas9/sgRNA expressing plasmids and the ssODN donor. Two days post-transfection, cells were replated, and transferred the next day to 39.5 °C. Two weeks later, colonies were stained with crystal violet and counted. Statistical analysis: Student’s t-test was performed, two-tailed, two-sample, unequal variance test. N = 7, *p < 0.02. (**C**,**D**). Reversion of TAF1 ts mutation in HEK293 is 2-fold more efficient with MRN-recruiting constructs. HEK293 TAF1 ts cells were transfected with the indicated Cas9/sgRNA-encoding plasmids and ssODN. Control non-targeting sgRNA and ssODN are listed in Appendix A. Cells were transfected and treated as in (B). C. Crystal violet-stained cells. D. Editing of HEK293 TAF1 ts by wt, U_N_, U_C,_ and U_NLS_ Cas9. Statistical analysis as in (B) N = 14, * *p* < 0.011.

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
