# Peer review of "Recruitment of DNA Repair MRN Complex by Intrinsically Disordered Protein Domain Fused to Cas9 Improves Efficiency of CRISPR-Mediated Genome Editing"

_biomolecules, 2019, doi:10.3390/biom9100584_

Round 1
Reviewer 1 Report
In the manuscript entitled “Recruitment of DNA repair MRN complex by intrinsically disordered protein domain fused to Cas9 improves efficiency of CRISPR-mediated genome editing”, the authors reported increased HDR efficiency using engineered Cas9 fused to intrinsically disordered domain of HSV-1 alkaline nuclease. Great efforts were made to optimize the effect of fused Cas9 on HDR rate mediated by both ssODN and dsDNA. However, I still have the following concerns to help improve the manuscript:
Major concerns:
The authors showed relative increase of HDR rate based on WB or flow cytometry results. However, no absolute or true HDR rate for WT and fused Cas9 was indicated all throughout the paper. As a study focusing on gene editing, these important information is necessary for its further application and shouldn’t be neglected. I recommend the authors determine the HDR rate by selecting single-cell colonies. Also, detailed flow cytometry results for each group should be shown. As described in the Introduction, fusion of Cas9 to CtIP, Mre11, or Rad52 showed improvement in HDR whereas reduction in NHEJ. However, the effect of Cas9-UL12 on HDR/NHEJ ratio was not explored. The genome editing efficiency largely depends on successful transfection of gene editing components, either plasmid or ribonucleoprotein complexes. Whenever evaluating the HDR rate, the observed HDR rate should be normalized according to the transfection efficiency. For the Figure 3H, the authors claimed that the cell number after puro selection was normalized to that without puromycin. In the UN and UC group, the values go over 1. I wondered why there is more cell number after puro selection? For the Figure 3J, there is different N numbers across groups. Are these data collected from a single WB. If not, these data points shouldn’t be pooled for analysis due to the variations in absolute ratio between blots. PCR-based detection methods are common and direct in identifying gene editing events. The authors could perform PCR in the 5’ and 3’ junction to demonstrate more directly that whether monoallelic or biallelic HDR occurred. Fusion of Cas9 to CtIP, Mre11, or Rad52 can be incorporated for comparison, which I believe would greatly increase the significance of this study .
Minor concerns:
Please clearly show the length of homology arm in each gene editing scenario. A few linguistic errors:
Line 24: “a simple means”
Line 178: ” Inrinsically”
Line 255: “We BHK21/13 ts13”
Line 298: “can result”
Author Response
We thank you for the review of our manuscript, and the constructive suggestions for how to improve it. We appreciate the opportunity to incorporate these suggestions into our revised manuscript, and feel that they have helped us to present our work more completely and clearly. Below please find our responses to the specific concerns raised (reviewer comments are in italics):
Reviewer 1:
Major concerns:
The authors showed relative increase of HDR rate based on WB or flow cytometry results. However, no absolute or true HDR rate for WT and fused Cas9 was indicated all throughout the paper. As a study focusing on gene editing, these important information is necessary for its further application and shouldn’t be neglected.
We now include the percentage of HDR-edited cells in the population for the PSMB6-YFP system as measured by FACS (Supplementary Table S2) and an estimate of the YFP-p73 HDR percentage, based on the XTT results (Supplementary Table S3).
I recommend the authors determine the HDR rate by selecting single-cell colonies.
HDR rate is often determined by the utilization of reporter based fluorescent genes. We however developed a more physiological strategy by targeting endogenous genes. The reviewer recommendation of analyzing selective single-cell colonies is indeed a reasonable way to go. However, in the experiment where we corrected a single point mutation (figure 4) the single-cell colonies we obtained only represent the correct edited cells and therefore we could not determine the non-HDR events. We know that mutation correction was more efficient with our constructs as compared to the naïve Cas9. Actually, we do not know the rate of NHEJ events using our strategy of recruiting the MRN complex.
For the other systems we used, such as PSMB6-YFP, FACS analysis indicated that the HDR success rate was only 1-2%. Therefore, analyzing single colonies would be laborious, and it would be difficult to obtain enough samples to be statistically significant. In addition, the NHEJ-repaired PSMB6 gene would probably be mutated, making those cells inviable, and skewing the results. We therefore used the FACS to evaluate HDR, which allowed us to screen thousands of cells per sample. We also used western blots to quantify and validate that the expected fusion protein was produced in each case, since only the correctly edited cells would express a protein of the expected size that was recognized by both the anti-YFP, and anti-PSMB6 antibodies. The same was true for the YFP-p73, and the PSMB6-Flag fusion products. In the case of the pac-2A-YFP-p73 cassette, properly edited cells should be puromycin resistant, and also express the YFP-p73 fusion protein. We showed that in this case the puromycin-resistant cells did also express the expected fusion protein, indicating faithful editing.
Also, detailed flow cytometry results for each group should be shown.
This data is now included in Supplementary Table S2.
As described in the Introduction, fusion of Cas9 to CtIP, Mre11, or Rad52 showed improvement in HDR whereas reduction in NHEJ. However, the effect of Cas9-UL12 on HDR/NHEJ ratio was not explored.
We developed strategies to obtain the highest number of HDR edited cells by preselecting the edited events. Because of the pre-selection process (as detailed above for correcting single point genetic mutation and inserting a DNA fragment) we could not evaluate the level of NHEJ rate.
We agree that it would be useful to know whether our chimeric constructs improve the HDR/NHEJ ratio in addition to improving the overall yield of HDR. We are planning in the future to test the Cas9 constructs in a system that is amenable to determining the HDR/NHEJ ratio, but those experiments will take time to complete.
The genome editing efficiency largely depends on successful transfection of gene editing components, either plasmid or ribonucleoprotein complexes. Whenever evaluating the HDR rate, the observed HDR rate should be normalized according to the transfection efficiency.
In each experiment we compared the pools of total transfected cells, which did include cells that might not have been transfected. Most of our experiments were performed in HEK293 cells, which normally have a high rate of transfection. However, it is true that transfection efficiency can vary, and low transfection efficiency would cause us to underestimate the absolute HDR rate, as determined, for example, by FACS. In any case, our primary interest was to compare the editing efficiency of the chimeric Cas9 constructs with the naïve Cas9. In each experiment, we monitored the levels of Cas9 expressed, which indicated equal transfection efficiency among the different samples we compared. In fact we observed that highest HDR events were obtained with much lower amounts of the Cas9 fusion protein as compared to the naïve Cas9. We also showed actin or tubulin levels to show total protein amounts loaded.
For the Figure 3H, the authors claimed that the cell number after puro selection was normalized to that without puromycin. In the UN and UC group, the values go over 1. I wondered why there is more cell number after puro selection?
We agree that the original version of the figure was not clear enough. We have now made some changes in the figure (Y-axis label) and legend to better explain the experimental strategy. We have also included Supplementary Table S3 with the raw data from the XTT assay. The cassette inserted encodes for the puromycin-resistance gene (puromycin acetyl transferase, pac), followed by a 2A peptide and YFP. Properly edited cells express pac and YFP-p73, and should be puromycin resistant. In order to quantify the amount of puromycin-resistant cells in the pools of transfected cells we used the XTT assay. Duplicate cell samples were plated with and without puromycin, and the XTT assay was used to quantify viable cells in each case.
For the Figure 3J, there is different N numbers across groups. Are these data collected from a single WB. If not, these data points shouldn’t be pooled for analysis due to the variations in absolute ratio between blots.
This figure summarizes the results of many western blots. In each blot, the band intensity achieved from one wt Cas9 sample was set as "1", and all the other bands were compared to this, giving a fold-change. This is why the wild type bar also has an error bar- since the replicates of the wild type Cas9 were also compared to the sample that was set as "1". These fold-change results from many independent western blot experiments were combined to produce the bar graph. Using the fold-change value allowed us to compare independent biological replicates, since the absolute intensity values of bands in western blots do vary from experiment to experiment. In the legend to the figure, we added the statement: "In each experiment, the level of PSMB6-YFP signal obtained with one of the wt Cas9 samples was set to "1", and the fold increase/decrease of the other samples was compared to this standard."
PCR-based detection methods are common and direct in identifying gene editing events. The authors could perform PCR in the 5’ and 3’ junction to demonstrate more directly that whether monoallelic or biallelic HDR occurred.
We have now included the PCR analysis of two clones of PSMB6-YFP edited cells (Figure S2C), both of which appear to be biallelic insertions of PSMB6-YFP. We have also included PCR analysis of the clones of edited cells expressing YFP-p73 (Supplementary Figure S4A) , whose western blot data is shown in Figure 3G.
Fusion of Cas9 to CtIP, Mre11, or Rad52 can be incorporated for comparison, which I believe would greatly increase the significance of this study.
We agree that the field might find interest in selecting the best strategy for achieving the highest number of HDR events, a likely outcome of comparing the different reported strategies with ours. But there are additional parameters to be considered. For example we show that with our strategy we reach the highest cases of HDR editing with much lower amounts of transfected DNA and proteins. Therefore, editing endogenous genes to optimize and to compare between the reported fusion proteins is not a trivial task.
This would be interesting to test, but cannot be performed within the time frame of the revision.
Minor concerns:
Please clearly show the length of homology arm in each gene editing scenario.
This information has been added.
A few linguistic errors:
Line 24: “a simple means”
Changed to "a simple approach to improve"
Line 178: ” Inrinsically”
Fixed
Line 255: “We BHK21/13 ts13”
Text changed to "We used BHK21/13 ts13"
Line 298: “can result”
Changed to "can occur"
Reviewer 2 Report
In the present study, Reuven et al. demonstrate how by fusing HSV recombinase domains get a significant increase in HDR. This is a key point in applying CRISPR system although it is an efficient system it does not perform very well in HDR experiments.
The data that the authors presented are very well explained and robust, probably the only experiment that I missed during the reading of the paper is the inclusion of Uc in the experiments in figure 2, due to the fact that the rest of the paper is very well supported by data with both fusions (Un and Uc). I would also recommend a careful revision of english language (minor modifications regarding spelling).
Author Response
We thank you for the review of our manuscript, and the constructive suggestions for how to improve it. Below please find our responses to the specific concerns (reviewer comments are in italics):
The data that the authors presented are very well explained and robust, probably the only experiment that I missed during the reading of the paper is the inclusion of Uc in the experiments in figure 2, due to the fact that the rest of the paper is very well supported by data with both fusions (Un and Uc).
The UN construct was the first construct we made, and the PSMB6-Flag system was one of the first we used. When we developed the PSMB6-YFP construct, we found it to be better for technical reasons (better detection of the YFP with our antibody, and fewer problems with cross-reacting bands). After making the UC construct, we found that both the UN and UC constructs performed similarly in the PSMB6-YFP, YFP-p73, and ts systems, but we did not repeat the PSMB6-Flag experiments with UC.
I would also recommend a careful revision of english language (minor modifications regarding spelling).
The manuscript was re-checked for spelling, and the language issues described above by reviewer 1 were fixed.
Round 2
Reviewer 1 Report
The authors have addressed most of my comments/suggestions. I found their responses quite satisfactory and the revised version has been much improved. For this reason I recommend the paper for publication in Biomolecules.